# Postoperative Seizure Prophylaxis in Meningioma Resection: A Systematic Review and Meta-Analysis

**DOI:** 10.3390/diagnostics13223415

**Published:** 2023-11-09

**Authors:** Sávio Batista, Raphael Bertani, Lucca B. Palavani, Leonardo de Barros Oliveira, Pedro Borges, Stefan W. Koester, Wellingson Silva Paiva

**Affiliations:** 1Faculty of Medicine, Federal University of Rio de Janeiro, Rio de Janeiro 21941-853, Brazil; 2Department of Neurosurgery, São Paulo University, Sao Paulo 05508-220, Brazil; neurosurgery@rbertani.com (R.B.);; 3Faculty of Medicine, Max Planck University Center, Indaiatuba 13343-060, Brazil; 4Faculty of Medicine, State University of Ponta Grossa, Ponta Grossa 84010-330, Brazil; 5Faculty of Medicine, Fundação Técnico-Educacional Souza Marques, Rio de Janeiro 21310-310, Brazil; pbborges9@gmail.com; 6School of Medicine, Vanderbilt University, Nashville, TN 37235, USA

**Keywords:** meningioma, seizure prophylaxis, antiepileptic drugs

## Abstract

Background: Seizures in the early postoperative period may impair patient recovery and increase the risk of complications. The aim of this study is to determine whether there is any advantage in postoperative seizure prophylaxis following meningioma resection. Methods: This systematic review was conducted in accordance with PRISMA guidelines. PUBMED, Web of Science, Embase, Science Direct, and Cochrane were searched for papers until April 2023. Results: Among nine studies, a total of 3249 patients were evaluated, of which 984 patients received antiepileptic drugs (AEDs). No significant difference was observed in the frequency of seizure events between patients who were treated with antiepileptic drugs (AEDs) and those who were not. (RR 1.22, 95% CI 0.66 to 2.40; I^2^ = 57%). Postoperative seizures occurred in 5% (95% CI: 1% to 9%) within the early time period (<7 days), and 9% (95% CI: 1% to 17%) in the late time period (>7 days), with significant heterogeneity between the studies (I^2^ = 91% and 97%, respectively). In seizure-naive patients, the rate of postoperative seizures was 2% (95% CI: 0% to 6%) in the early period and increased to 6% (95% CI: 0% to 15%) in the late period. High heterogeneity led to the use of random-effects models in all analyses. Conclusions: The current evidence does not provide sufficient support for the effectiveness of prophylactic AED medications in preventing postoperative seizures in patients undergoing meningioma resection. This underscores the importance of considering diagnostic criteria and conducting individual patient analysis to guide clinical decision-making in this context.

## 1. Introduction

Meningiomas, the most common primary intracranial tumors, arise from arachnoid meningothelial cells and are typically diagnosed using contrast-enhanced magnetic resonance imaging for its superior soft tissue imaging capabilities and safety due to the absence of radiation exposure [1]. These tumors are classified as extra-axial, meaning they develop outside the brain parenchyma [2]. On imaging, meningiomas are often visualized as well-defined sessile or lentiform lesions with broad-based dural attachments. Additional radiological features such as dural tails, hyperostosis in the underlying bones, linear internal flow voids, and calcification can provide important clues for the diagnosis of these tumors. Benign meningiomas primarily receive their blood supply from dural branches of the external carotid artery, resulting in strong and uniform enhancement following the administration of contrast agents [2,3]. Peritumoral edema, observed in up to 60% of cases, is a significant risk factor for the occurrence of seizures [4].

Despite being generally classified as benign, meningiomas can cause significant neurological symptoms and complications, including seizures [5,6]. Seizures occur in approximately 15–30% of patients with meningiomas [5]. Surgical resection is the primary treatment approach, aimed at alleviating seizures by removing the tumor and relieving pressure on the surrounding brain tissue. Although the majority of patients experience seizure freedom after surgery (60–90%), it may persist in 30–40% of patients with a history of seizures and 10–15% of patients without a seizure history [7]. Therefore, the risk of postoperative seizures, particularly in the immediate postoperative period, remains a concern [6]. To address this, postoperative seizure prophylaxis plays a crucial role in the management of patients undergoing meningioma resection.

Postoperative seizure prophylaxis involves the administration of antiepileptic drugs (AEDs) with the aim of preventing seizures following tumor resection. After tumor removal, the brain undergoes a period of adjustment and healing, during which abnormal electrical activity can occur, potentially leading to seizures [7]. AEDs are commonly used to prevent or reduce this abnormal activity by modulating the electrical signals in the brain. Different AEDs work through various mechanisms of action depending on the specific drug.

One mechanism of action involves stabilizing neuronal membranes. AEDs, such as phenytoin and carbamazepine, block sodium channels in neurons, preventing the rapid and excessive flow of sodium ions that can trigger uncontrolled electrical activity and seizures [8,9]. By stabilizing the neuronal membranes, these drugs help maintain the normal balance of electrical signals in the brain. Another mechanism involves modulating neurotransmitter activity. AEDs, such as levetiracetam and valproic acid, act on neurotransmitters like gamma-aminobutyric acid (GABA), an inhibitory neurotransmitter that helps regulate neuronal excitability [10,11]. These drugs increase the concentration or enhance the activity of GABA, reducing the likelihood of hyperexcitability and seizure activity.

By administering AEDs following tumor resection, therapeutic levels of the medication are established in the body, providing ongoing protection against abnormal electrical activity and reducing the risk of postoperative seizures. The choice of AEDs may depend on factors such as the patient’s medical history, tumor characteristics, and potential drug interactions. The dosage and duration of AED treatment may also vary based on individual patient characteristics and the recommendations of the surgical team. It is important to note that while AEDs are effective in preventing seizures, they are not without side effects. Some individuals may experience adverse reactions, such as drowsiness, dizziness, or cognitive changes [12].

The primary goal of seizure prophylaxis is to minimize the risk of early postoperative seizures, which can have detrimental effects on patient recovery and increase the risk of complications. Early seizures can also disrupt the healing process, impair neurological function, and potentially prolong hospital stays [4]. This translates into delayed initiation and overall duration of rehabilitation, as well as an increased risk of hospital infections. Early postoperative seizures were also associated with novel neurological deficits and aspiration pneumonia [13].

The decision to initiate seizure prophylaxis is based on several factors, including the patient’s pre-operative seizure history, tumor location and characteristics, and individual risk factors. Patients with a history of pre-operative seizures, large tumors, cortical involvement, or tumors located in eloquent areas of the brain may be at higher risk for postoperative seizures [14,15]. However, the optimal duration and necessity of seizure prophylaxis remain a subject of debate and may vary depending on individual patient characteristics.

In this paper, we conducted a comprehensive systematic review and meta-analysis of studies investigating the prophylactic use of AEDs in the postoperative period following meningioma resection. Our aim was to synthesize the available evidence and provide a comprehensive analysis of the effectiveness of AED prophylaxis in this patient population.

## 2. Materials and Methods

### 2.1. Eligibility Criteria

This comprehensive meta-analysis incorporated studies focusing on seizure prophylaxis in surgically resected meningiomas. Studies were excluded from our analysis if they lacked reported data on seizure events. Additionally, we excluded case reports, letters, comments, and reviews to focus solely on studies that provided substantive and comprehensive data. Furthermore, to enhance the quality and relevance of the study, we excluded research papers that did not provide clear data on whether they were reporting meningioma tumors or not. This exclusion was implemented to ensure that only high-quality studies were included in the analysis.

This study examined the postoperative prophylaxis in both the early and late stages following meningioma resection. The early stage was defined as the time period up to 7 days postoperatively, while the late stage encompassed studies with follow-up ranging from more than 7 days up to 12 months. Additionally, a comparison was conducted between patients who had previously experienced seizures and those who developed new-onset seizures. This comprehensive analysis aims to provide insights into the efficacy of prophylactic measures across different time periods and patient subgroups.

### 2.2. Search Strategy and Risk of Bias Assessment

A comprehensive systematic search was conducted across multiple databases, including PUBMED, Science Direct, Cochrane, Embase, and Web of Science. The search strategy employed the following terms: “seizures”, “epilepsy”, “prophylaxis”, “prophylactic”, “prevention”, “preventive”, “antiepileptic”, “anticonvulsant”, “phenytoin”, “levetiracetam”, “valproic”, “carbamazepine”, “gabapentin”, and “meningioma”.

Two authors (P.B. and L.P.) independently extracted the data, adhering to predefined search criteria and quality assessment guidelines. The extracted data were subsequently reviewed by two additional authors (S.B. and R.B.) to ensure accuracy and consistency. Search was registered with a PROSPERO ID [CRD42023438015].

The methodological bias assessments of the included studies were performed in accordance with the risk of bias in non-randomized studies of interventions (ROBINS-I) tool [16] by two authors (P.B. and S.B.). This tool was used to assess the potential bias in the study designs, conduct, and reporting of the included studies. The application of the ROBINS-I tool allows for a standardized and systematic evaluation of the methodological quality of non-randomized intervention studies, providing a robust assessment of potential biases in the analyzed data.

### 2.3. Statistical Analysis

This systematic review and meta-analysis was performed in accordance with the Cochrane Collaboration and the Preferred Reporting Items for Systematic Reviews and Meta-Analysis (PRISMA) statement guidelines [17]. Pooled analysis of the studies with 95% confidence intervals was used to compare AED prophylaxis effects for outcomes. I^2^ statistics were used to assess for heterogeneity; p-values inferior to 0.05 and I^2^ < 35% were considered significant for heterogeneity, and in these cases, a random-effects analysis was performed, instead of fixed-effects. The whole statistical analysis was performed using the software R (version 4.2.3, R Foundation for Statistical Computing, Vienna, Austria), and Review Manager (version 5.4.1).

## 3. Results

### 3.1. Study Selection

A comprehensive search of the multiple databases yielded a total of 5284 articles. After removing duplicates, 3076 unique citations were screened. Following a thorough review of titles and abstracts, 3014 articles were excluded. Of the remaining articles, 62 were selected for full-text review based on their abstracts. Subsequently, a further 53 articles were excluded after the full-text screening and data extraction process. Ultimately, nine studies met the inclusion criteria and were included in the final analysis [18,19,20,21,22,23,24,25,26]. A detailed overview of the search process can be found in Figure 1.

### 3.2. Quality of the Included Studies

The assessments of methodological bias risk were visually presented in Figure 2, offering a comprehensive overview of the included studies. Among the evaluated studies, eight were deemed to have an overall moderate risk of bias based on the assessment of bias domains, while one study was determined to have a serious risk of bias.

To further illustrate the contributions of bias within the network, Figure 3 depicts the average risk of bias for each comparison, segregated by each criterion and the overall risk of bias for the included studies. This provides valuable insights into the potential impact of bias on the findings for the included studies, aiding in the interpretation of the overall results.

### 3.3. Baseline Characteristics of Included Studies

A total of 3249 patients from nine retrospective studies conducted between 1996 and 2019 were included in the analysis. The average age of the patients at diagnosis was 55.2 years. The distribution of patients by sex showed a significant difference, with 969 males (29.8%) and 2280 females (70.2%) included in the studies. The drugs used for seizure prophylaxis varied across the studies, without an analysis of the effect of each drug individually. Overall, phenytoin was present in five out of nine studies, levetiracetam in four out of nine studies, carbamazepine in two out of nine studies, and valproic acid in one out of nine studies.

Table 1 summarizes the main characteristics of patients in each study. The localization of meningiomas was reported in 2772 out of 3249 patients, with 1229 located in the skull base (44.3%) and 1543 in non-skull base regions (55.6%). Besides skull base and non-skull base division, 227 were parasagittal, and 99 were distributed across the frontal, parietal, occipital, and temporal regions. The tumor classification according to the World Health Organization (WHO) grade showed that grade I was the most prevalent in 2232 cases (77.7%), followed by grade II in 567 cases (19.7%), and grade III in 70 cases (2.4%). The surgical resection and Simpson grade (SG) evaluation reported 1532 (91%) cases of gross total resection (GTR) and 150 (8.9%) cases of subtotal resection (STR). The SG were classified as I–III in 1795 cases (87%) and IV–V in 268 cases (12.9%).

Preoperative seizures were reported in seven out of nine studies, totaling 574 patients, while patients without seizures were reported in eight out of nine studies, totaling 2327 patients. The Wirsching 2016 study [24] examined a total of 661 locations, specifically excluding data on multiple meningiomas. This deliberate exclusion explains the difference of 118 reports from the total of 779 patients reported. The size of the tumors was described using centimeters (cm), millimeters (mm), and cubic centimeters (cm^3^). Among the included studies, four used cm as the unit of measurement, two used cm^3^, and only one used mm.

### 3.4. Postoperative Seizures in AEDs and No AEDs

A comparison between patients who received AEDs and those who did not was conducted in six out of nine studies, involving a total of 2202 patients. The analysis revealed a risk ratio of 1.22 for postoperative seizure events, with a 95% confidence interval ranging from 0.66 to 2.40, using a random-effects model (Figure 4). This value indicates a lack of significant difference in seizure incidence between the two groups. The presence of significant heterogeneity among the studies (I^2^ = 57%) led to the preference for conducting the analysis using a random-effects model.

### 3.5. Postoperative Seizures Prophylaxis in Early and Late Time

In a pooled analysis using a random-effects model, six out of nine studies involving 1043 patients who were prescribed AEDs were analyzed to assess the occurrence of postoperative seizures in the early time period (<7 days). The pooled analysis revealed a seizure rate of 5% (95% CI: 1% to 9%) in this group, in a random-effects model due to a high level of heterogeneity between the studies (I^2^ = 91%) (Figure 5).

Similarly, in the late time period (>7 days), the rate of postoperative seizures was found to be 9% (95% CI: 1% to 17%), also analyzed using a random-effects model, and demonstrating substantial heterogeneity (I^2^ = 97%) (Figure 6). These findings suggest that there is a higher incidence of seizures in the late postoperative period compared to the early postoperative period, indicating a natural history of seizures in these two times. Due to the absence of comparative data between patients who did not receive AEDs, it is not possible to definitively establish a potential protective effect of AEDs during the early stage.

### 3.6. New Onset Postoperative Seizures

In a comprehensive analysis of four studies involving 743 patients who did not have pre-operative seizures but developed them only after the meningioma resection, a total of 41 patients experienced seizures after resection in the early time period, while 60 patients experienced seizures in the late time period. The analysis revealed that the rate of postoperative seizures in the early time period was 2% (95% CI: 0% to 6%), as seen in Figure 7. Among these studies, only the study by Li et al. reported cases of new-onset seizures, involving a total of 41 patients.

However, in the late time period, the rate of postoperative seizures increased to 6% (95% CI: 0% to 15%), as shown in Figure 8. Due to the substantial heterogeneity observed among the studies (I^2^ = 92% and 95%, respectively), both analyses were performed using random-effects models. The data suggests that new-onset seizures occurring in the late time period after surgical resection are more susceptible to seizure events compared to the early time period. However, similar to the previous findings, the absence of comparative data between patients who did not receive AEDs prevents us from definitively establishing a potential protective effect of AEDs during the early stage in patients with new-onset seizures postoperatively.

## 4. Discussion

The primary objective of this study was to determine whether to or not perform seizure prophylaxis following surgical resection of meningiomas. To address this question, we conducted a comprehensive systematic review and meta-analysis of the available literature until April 2023, focusing on the utilization of AEDs post-meningioma resection. The analysis involved assessing the frequency of seizures at various time points after the surgical procedure in a sample of 3249 patients with meningiomas who underwent resection. Among these patients, 1043 individuals received seizure prophylaxis. A comprehensive binary comparison was conducted to analyze the differences between patients who received AEDs and those who did not overall, and single-arm analysis specifically in terms of the rates of new-onset seizures and non-new seizures during both the early and late stages of seizure occurrence. Upon conducting a thorough examination of the data, the study’s findings indicate that administering anticonvulsant prophylaxis to meningioma patients without seizures is not universally justified, as previously reported by Delgado Lopez et al. [27] and Englot et al. [6]. Instead, our study underscores the importance of specific pre-surgical risk factors, such as diagnostic criteria, in the development of postoperative seizures.

Our findings indicate that the use of AEDs for seizure prophylaxis may not be optimal for the majority of patients. After surgery, the frequency of seizures was not significantly reduced in patients treated with AEDs and, in some cases, it was even higher. This seemingly paradoxical outcome can be attributed to the nature of some studies conducted, where observational studies tend to allocate AEDs to patients with more severe conditions [27]. Additionally, it is important to consider that AEDs are not devoid of risks, as they can lead to various adverse effects depending on the specific medication used. Common adverse effects include drowsiness and fatigue, dizziness and coordination issues, cognitive and memory impairments, mood changes and behavioral alterations, skin rashes and hypersensitivity reactions, weight fluctuations, gastrointestinal problems, liver complications, blood-related side effects, and potential teratogenic effects [28,29,30]. Additionally, after meningioma resection, up to half of patients may experience drug-related side effects, which can negatively impact their quality of life and neurocognitive function [19]. To further illustrate this point, a study by Tanti MJ et al. [31] demonstrated the use of AEDs to have a greater impact on the quality of life of patients with meningiomas than recent seizures.

Furthermore, it is noteworthy that AEDs are not cost-free, and patients worldwide may not consistently adhere to their prescribed treatment [32]. In fact, Faught et al. [33]. reported that each non-adherent patient incurs an additional cost of 4623 USD per quarter compared to adherent patients. Sughrue et al. [22] bring this matter into their discussion, questioning whether the benefit of AEDs after meningioma resection is worth the costs and side effects these medications carry. This highlights the potential iatrogenic effects and unnecessary financial burden that can result from the inappropriate use of AEDs when not warranted.

The effectiveness of antiepileptic prophylaxis following meningioma resection remains a topic of debate and varying perspectives. Our study findings indicate some conflicting results in this regard. Yang et al. conducted a retrospective analysis of 186 cases and found no reduction in the incidence of perioperative seizures among patients who received antiepileptic prophylaxis for 7 continuous days post-surgery [34]. Similarly, Sughrue et al. observed no significant difference in seizure rates between patients receiving antiepileptic drugs (129 patients) and those who did not (51 patients) after meningioma resection surgery [22]. However, Wang et al. argue that antiepileptic drugs are necessary for patients with atypical and malignant meningiomas [23]. In line with these findings, Islim et al. suggest a more targeted approach for the use of antiepileptic drugs, taking into consideration specific risk factors [19]. These divergent outcomes highlight the complexity of determining the optimal use of antiepileptic prophylaxis.

In order to comprehensively understand the indications for AEDs following meningioma resection, it is important to recognize the significant burden that epilepsy imposes on patients in terms of both morbidity and mortality [19]. As a result, some authors advocate for prophylactic measures in all patients, even in the absence of substantial evidence supporting their use [35,36]. However, our study emphasizes the need for a rational and individualized approach when evaluating the risk factors associated with postoperative seizures. These risk factors include tumor size, the extent of cortical involvement, specific radiological and histopathological characteristics such as alterations in resonance sequences, irregular tumor shape, absence of dural tail, presence of preoperative seizures, histological grade, the extent of tumor resection, and the biological behavior of any remaining tumor remnants [19,20,21,24,25]. The decision regarding the need for prophylaxis or additional treatment should be based on a careful assessment of these risk factors through diagnostic tools [23], because by taking into account these factors, clinicians can tailor their approach to seizure management on an individual basis, optimizing patient outcomes.

The location of the lesion has been identified as a significant risk factor. Islim et al. demonstrated a higher risk of postoperative seizures in patients with lesions located at the convexity and fronto-parietal regions [19]. Similarly, other studies have indicated an increased risk of postoperative seizures when lesions are not located in the skull base [21,25]. Chozick et al. similarly highlighted the higher attention demand associated with parietal locations [18]. These findings emphasize the importance of considering the specific location of the lesion when assessing the risk of postoperative seizures. Taking into consideration the necessity to evaluate individual patients’ characteristics before the use of AEDs, making a radiological diagnosis, and noticing possible risk factors associated with a location, the responsible physician can decide the possibility to intervene using AEDs and preventing seizure episodes.

In terms of seizure timing, our findings reveal a higher incidence of seizures occurring after 7 days following the surgical procedure. This aligns with the observations made by Wang et al., who suggest that AEDs are effective in preventing early seizures but may have limited efficacy in preventing late events [23]. Xue et al. even raise the possibility of AED withdrawal after one week of surgery in the absence of seizures [25]. Therefore, it is crucial to consider the optimal duration for AED use. Furthermore, providing ongoing monitoring and support to patients for several months following the procedure is essential to promptly detect and manage any potential seizure episodes.

## 5. Limitations

It is important to address the limitations of our study. The inclusion of only retrospective studies introduces potential biases, most notably selection bias. However, we took steps to address these concerns by conducting a thorough quality assessment of the included studies. As mentioned earlier, several studies have also acknowledged the presence of selection bias. In addition, the potential for underestimating seizure incidence during the postoperative follow-up period was recognized as a limitation in nearly all the studies. We acknowledge this as a potential limitation in our own study as well. Additionally, the average score of the studies included in our meta-analysis suggests an intermediate level of quality. It is worth noting that our results exhibited a high degree of heterogeneity. Considering the lack of studies describing tumor type and seizure characteristics hinders a thorough examination of potential connections. Subsequent studies ought to conduct comprehensive histopathological analysis for each tumor, enabling more precise comparisons and facilitating an analysis of the necessity for using AEDs. The absence of studies in the literature that document the dosage regimens of AEDs and provide detailed patient profiles poses challenges in assessing the true effectiveness of each AED available. Lastly, the studies present in the literature report a variable long-term follow-up, which makes the inconsistent data for months or even years difficult to make homogeneous comparisons. We suggest that future studies report more detail on the period considered “late”, as it may be useful to investigate the time frame between the use of AEDs and the development of seizures. Finally, to gain a more comprehensive understanding, future randomized clinical trials should be conducted to shed further light on these subjects, especially taking into consideration the effects of tumor size and location. In the meantime, the prescription of AEDs should be tailored to each individual based on their meningioma diagnosis, ensuring a personalized approach.

## 6. Conclusions

These findings strongly emphasize the importance of a cautious and individualized approach when considering the use of AEDs for all patients. The study revealed that there was no significant reduction in seizure frequency and, in some cases, even a potential increase. These results emphasize the criticality of conducting a thorough diagnosis, taking into account various risk factors, such as tumor size, cortical expression, radiological characteristics, preoperative seizures, histological grade, the extent of resection, and tumor behavior. It is crucial to carefully evaluate these factors to determine the need for prophylaxis or any supplementary treatments. Furthermore, it is essential to take into account the potential adverse effects of AEDs and the financial implications associated with non-adherence to these medications. Additionally, it is crucial to provide timely monitoring and support following the procedure, as the majority of seizures tend to occur within 7 days post-operation. The literature is notably deficient in studies that examine the correlation between resection grade, tumor type, postoperative seizure occurrence, and the effectiveness of prophylactic measures. The absence of such studies limits our understanding of how these factors interplay and their impact on postoperative seizure outcomes. Future investigations should prioritize comprehensive analyses that include the assessment of resection grade, tumor type and the implementation of prophylactic measures. Deepening into these aspects, researchers can provide valuable insights into the optimal management strategies for postoperative seizure prophylaxis, ultimately improving patient care and outcomes.

## Figures and Tables

**Figure 1 diagnostics-13-03415-f001:**
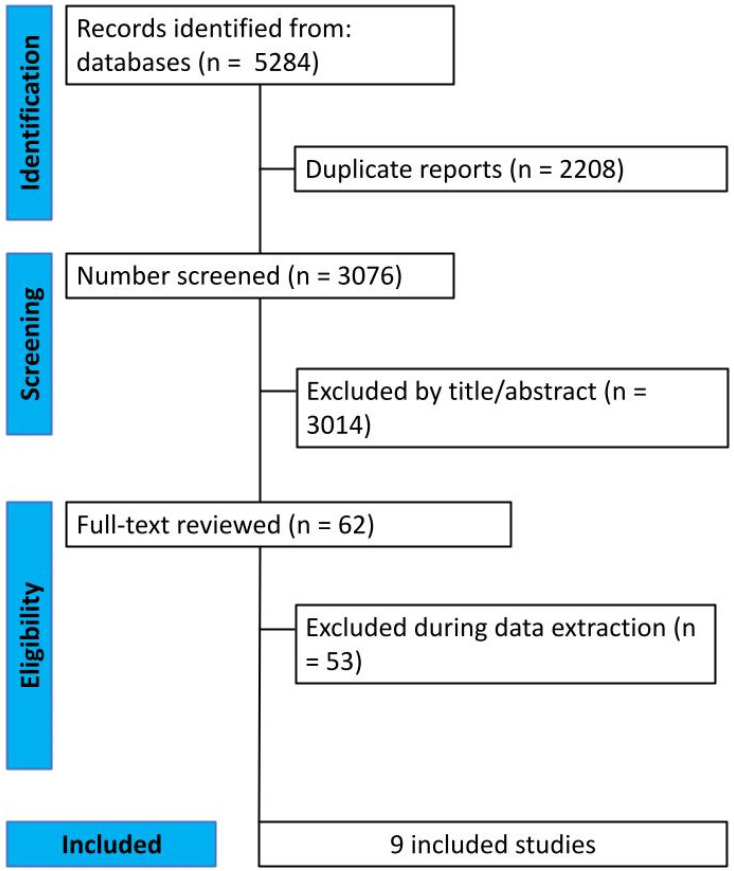
PRISMA Flow Diagram.

**Figure 2 diagnostics-13-03415-f002:**
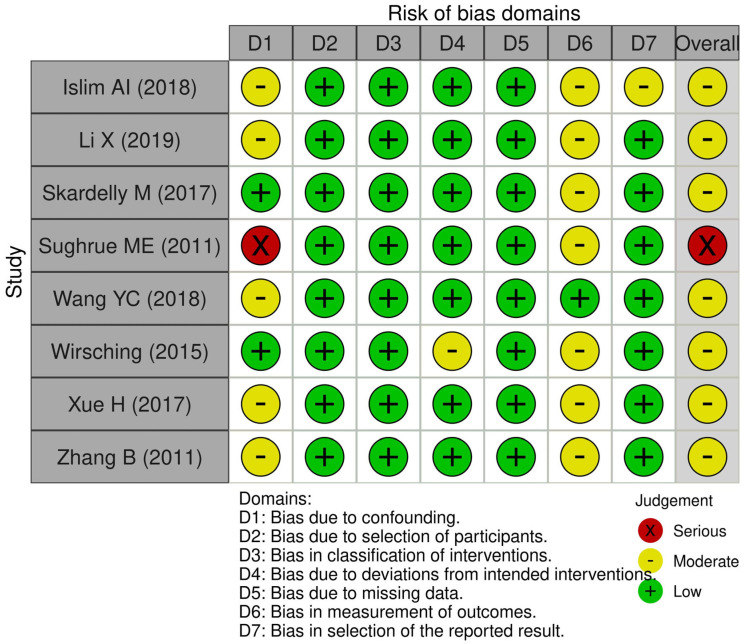
ROBINS-I tool for risk of bias assessment [19,20,21,22,23,24,25,26].

**Figure 3 diagnostics-13-03415-f003:**
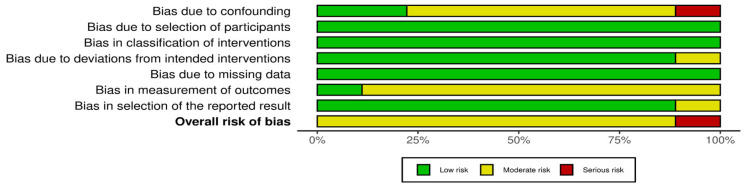
Average risk of bias contributions.

**Figure 4 diagnostics-13-03415-f004:**
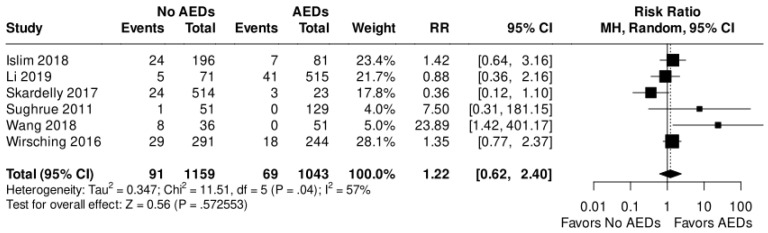
Postoperative seizure incidence comparison between AEDs and no AEDs groups [19,20,21,22,23,24].

**Figure 5 diagnostics-13-03415-f005:**
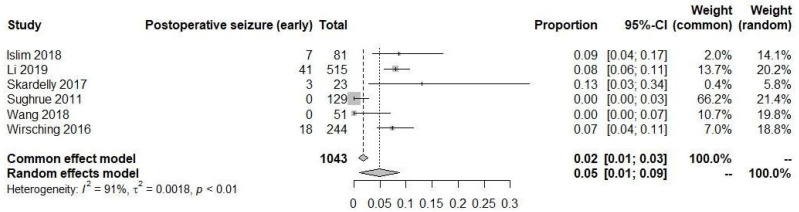
Postoperative seizure in early time [19,20,21,22,23,24].

**Figure 6 diagnostics-13-03415-f006:**
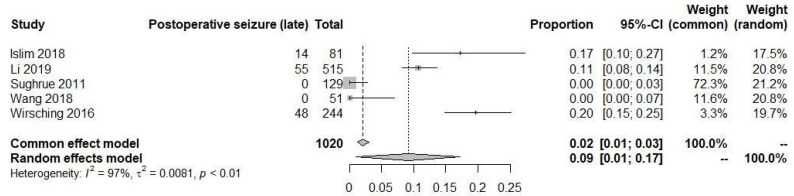
Postoperative seizure in late time [19,20,22,23,24].

**Figure 7 diagnostics-13-03415-f007:**
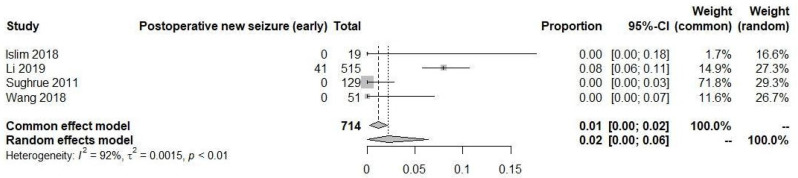
Postoperative new-onset seizure in early time [19,20,22,23].

**Figure 8 diagnostics-13-03415-f008:**
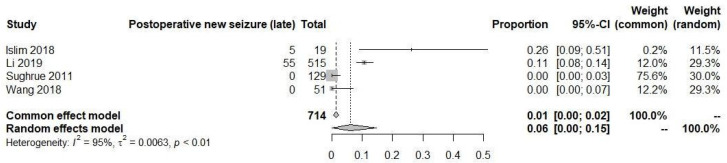
Postoperative new-onset seizure in late time [19,20,22,23].

**Table 1 diagnostics-13-03415-t001:** Main characteristics of included patients.

Study	*n*	Mean Age	Sex (M/F)	Drugs	Preoperative Seizures	Patients without Preoperative Seizures	Size of Tumor	Tumor Location	WHO Grade	SG/Resection
Chozick, 1996 [18]	158	60	50/108	NA	63	95	NA	Skull base: 141 * (Frontal 38 (26.9%), Parietal 34 (24.1%), Middle cranial fossa 12 (8.5%), Occipital 6 (4.2%), Sphenoid wing 33 (23.4%), Petrous 8 (5.6%), Dorsum/tuberculum sella 7 (4.9%), Cavernous sinus 3 (2.1%)) Non-skull base: 60 * (Parasagittal 56 (93.3%), Intraventricular 4 (6.6%))	NA	SG NA GTR 119; STR 39
Islim, 2018 [19]	283	57	69/214	P 48%, L 26%	68	215	≤10 cm^3^ 57 > 10 cm^3^ 221	Skull base: 76 (Sphenoid 34 (44.7%), Olfactory groove 18 (23.6%), Suprasellar 10 (13.1%), Posterior fossa 2 (2.6%), Others 12 (15.7%)) Non-skull base: 207 (Convexity 98 (47.3%), Parafalcine 39 (18.8%), Tentorial 24 (11.5%), Convexity/parafalcine 17 (8.2%), Parasagittal 12 (5.7%), Posterior fossa 5 (2.4%), Others 12 (5.7%))	WHO GRADE I 233, II 47, III 3	NA
Li, 2019 [20]	778	50	241/537	NA	87	586	Range 1.2–11.5 cm	Skull base: 291 Non-skull base: 487	WHO GRADE I 699, II 74, III 5	SG I-III 735 SG IV 43 GTR 735 STR 43
Skardelly, 2017 [21]	634	58	176/458	L: 76, other: 30, missing: 3/L: (median/10–90/range)—1000 mg/1000–2000 mg/500–3000 mg	97	537	Tumor volume (Median) 17.4 cm^3^ skull base meningioma 10.8 cm^3^ cranial roof meningioma 26.4 cm^3^	Skull base: 350 Non-skull base: 284	WHO GRADE I 423, II 208, III 3	SG I 142 II 83 III 181 IV 115 Missing 16. Resection: NA
Sughrue, 2011 [22]	180	55	50/130	P and L	0	180	NA	NA	WHO GRADE I 129, II 30, III 21	SG I 102 II 59 III 10 IV 9. Resection: NA
Wang, 2018 [23]	102	57	45/57	Va, L or P	15	87	Tumor Diameter (cm) Mean. 5.2 ± 2.0	Skull base: 29 Non-skull base: 73 (convexity 33 (45.2%), parasagittal 32 (43.8%), posterior fossa 8 (10.9%))	WHO GRADE II 86, III 16	SG NA GTR 69 STR 33
Wirsching, 2016 [24]	779	57	247/532	P and CB	244	535	Maximal diameter (mm) Median 40.0	Skull base: 260 Non-skull base: 401 * (convexity 167 (41.6%), parasagittal 131 (32.6%), posterior fossa 81 (20.1%), other location 22 (5.4%))	WHO GRADE I 638, II 119, III 22	SG I 143 II 221 III 41 IV 55 V11 GTR was achieved in 531 patients. A subset of 129 patients underwent 2 or more meningiomas
Xue, 2018 [25]	113	53	19/94	CB	NA	92	<3.5 cm (65) and ≥3.5 cm (50)//median (range): 3.51 ± 1.58 cm	Skull base: 82 (spenhoid wing 25 (30.4%), posterior fossa 19 (23.1%), middle cranial fossa 9 (10.9%), olfactory groove 8 (9.7%), tuberculum sellae 8 (9.7%), tentorial 7 (8.5%), foramen magnun 5 (6.09%), foramen jugular 1 (1.2%)) Non-skull base: 31 (convexity 17 (54.8%), parasaggital 8 (25.8%), falcine 4 (12.9%), intra-lateral ventricle 2 (6.4%))	WHO GRADE I 110, II 3	SG I–III 78, IV–V 35 GTR 78 STR 35
Zhang, 2011 [26]	222	50	72/150	P 5 mg/kg per day	NA	NA	Maximum tumour diameter—(1) 3–5 cm; (2) <3 cm; and (3) >5 cm	Skull base and non-skull base: * Spine of sphenoid bone, parietooccipital, posterior fossa, occipitotemporal area, occiput, cerebellopontine angle, cerebellar hemisphere, frontotemporal, middle cranial fossa, petroclival, paracele, anterior cranial fossa, forehead, saddle region, and temporoparietal region, temporal, parietal lobe, frontal and parietal lobes	NA	NA

* Disagreement between location reported and total of patients. Male (M), Female (F), Phenytoin (P), Levetiracetam (L), Carbamazepine (CB), Simpson grade (SG), Gross total resection (GTR), subtotal resection (STR), Not available (NA).

## Data Availability

Not applicable.

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
