# Peer review of "Postoperative Seizure Prophylaxis in Meningioma Resection: A Systematic Review and Meta-Analysis"

_diagnostics, 2023, doi:10.3390/diagnostics13223415_

Round 1

Reviewer 1 Report

This systematic review evaluates the benefit of postoperative seizure prophylaxis following meningioma resection. The paper is well-crafted and the study thoroughly conducted. I have two points for consideration:

1. The authors assert that prophylactic AEDs do not effectively prevent postoperative seizures in meningioma patients, but various factors may impact these results, such as tumor size or location. Could the authors consider controlling one of these variables while examining prophylactic AED effects? For example, could they categorize patients with similar tumor sizes or locations from different studies and evaluate the effects within these groups?

2. Some previously published studies have explored the effects of prophylactic AED. What novel insights does this study offer in comparison to existing research?

Author Response

This systematic review evaluates the benefit of postoperative seizure prophylaxis following meningioma resection. The paper is well-crafted and the study thoroughly conducted. I have two points for consideration:

#1 The authors assert that prophylactic AEDs do not effectively prevent postoperative seizures in meningioma patients, but various factors may impact these results, such as tumor size or location. Could the authors consider controlling one of these variables while examining prophylactic AED effects? For example, could they categorize patients with similar tumor sizes or locations from different studies and evaluate the effects within these groups?

Response:

We sincerely appreciate the reviewer's valuable input. Controlling for variables like tumor size or location could indeed provide further insights into the effects of prophylactic AEDs on postoperative seizures in meningioma patients. In our systematic review, we acknowledge the limitations of the available data, including the lack of studies describing tumor type and seizure characteristics, which hindered a thorough examination of potential connections. We have added a suggestion in the limitations section of our paper to encourage future research that categorizes patients with similar tumor sizes or locations from different studies and evaluates the effects within these groups. This approach could help uncover potential relationships between specific patient characteristics and the effectiveness of prophylactic AEDs in preventing postoperative seizures.

#2 Some previously published studies have explored the effects of prophylactic AED. What novel insights does this study offer in comparison to existing research?

Response: 

We appreciate the opportunity to highlight the novel insights our study offers in comparison to existing research. While previous studies such as Delgado Lopez et al. (Neurologia 2023), Englot et al. (J Neurosurgery 2016), and Sue et al. (Acta Neurochir 2015) have examined the influence of AEDs on postoperative seizures in meningioma patients, our study builds upon these findings in several key ways:

  1. Our analysis includes seven new studies that were not part of the investigations conducted by Sue et al. and Englot et al., thereby expanding the scope of data available for analysis.

  1. We performed a pooled analysis of data, allowing us to provide a more comprehensive overview of the incidence of seizure events in the total patient population. This pooled analysis provides a more robust assessment of the effects of prophylactic AEDs on postoperative seizures in meningioma patients.

iii. We distinguished between early (within 7 days) and late (>7 days) time periods to gain insight into the timing of seizure occurrences. This distinction is particularly relevant, as it aligns with the findings of Wang et al. (2018), who suggested that AEDs might be more effective in preventing early seizures but may have limited efficacy in preventing late events in surgically resected atypical and malignant meningiomas. By analyzing and presenting data separately for these time periods, we contribute to a better understanding of the temporal dynamics of postoperative seizures in relation to prophylactic AED use.

We hope that by providing these additional insights, our systematic review contributes to the existing knowledge on the benefits of postoperative seizure prophylaxis following meningioma resection. We thank the reviewer for their thoughtful comments, which have helped strengthen the impact and significance of our study.

Reviewer 2 Report

I have a few comments:

1. The Abstract leaves the reader somewhat confused with regards to the main findings.

You write that "A comparison between patients who received AEDs and those who did not showed a non-significant differences between both groups (RR 1.22, 95% CI 0.66 to 2.40; I² = 57%)."

What parameter did you compare?

You also state that "Postoperative seizures occurred in 5% (95% CI: 1% to 9%) within the early time period (<7 days), and 9% (95% CI: 1% to 22 17%) in the late time period (>7 days), with significant heterogeneity (I² = 91% and 97% respectively). 

Heterogeneity of what? You should make this more clear.

2. It is not clear what the "main outcome" you refer to actually is (Methods section, lines 111-114).

3. Your exclusion criteria are somewhat vague - what is "clear data on meningioma tumors" (Methods section)?

4. Your Introduction and Discussion should account for the fact that one of your main research questions has been investigated before. For example,  Delgado Lopez et al (Neurologia 2023) as well as Englot et al (J Neurosurgery 2016) and Sue et al (Acta Neurochir 2015) all found that AEDs do not appear to influence postoperative seizures in meningioma. How does your study compare to those, and what novel aspects does yours add?

There are multiple language issues, e.g. 

"Among 9 studies, a total of 3,249 were evaluated patients..." (lines 18, 19)

"...showed a non-significant differences..." (lines 20, 21)

Author Response

#1 The Abstract leaves the reader somewhat confused with regards to the main findings.

You write that "A comparison between patients who received AEDs and those who did not showed a non-significant differences between both groups (RR 1.22, 95% CI 0.66 to 2.40; I² = 57%)."

What parameter did you compare?

You also state that "Postoperative seizures occurred in 5% (95% CI: 1% to 9%) within the early time period (<7 days), and 9% (95% CI: 1% to 22 17%) in the late time period (>7 days), with significant heterogeneity (I² = 91% and 97% respectively). 

Heterogeneity of what? You should make this more clear.

Response: 

Thank you for pointing out the confusion in the abstract. We apologize for the oversight. The main parameter we compared in the study was the occurrence of postoperative seizures between patients who received AEDs and those who did not.

We have switched the previous version “A comparison between patients who received AEDs and those who did not showed a non-significant differences between both groups rega (RR 1.22, 95% CI 0.66 to 2.40; I² = 57%).” to this “A comparison between patients who received AEDs and those who did not showed a significant difference between both groups regarding seizure events (RR 1.22, 95% CI 0.66 to 2.40; I² = 57%)” 

With respect to heterogeneity, it was not clear, thus we have updated from this “Postoperative seizures occurred in 5% (95% CI: 1% to 9%) within the early time period (<7 days), and 9% (95% CI: 1% to 17%) in the late time period (>7 days), with significant heterogeneity (I² = 91% and 97% respectively).” to this “Postoperative seizures occurred in 5% (95% CI: 1% to 9%) within the early time period (<7 days), and 9% (95% CI: 1% to 17%) in the late time period (>7 days), with significant heterogeneity between the studies (I² = 91% and 97% respectively).”

#2 It is not clear what the "main outcome" you refer to actually is (Methods section, lines 111-114)

Response: 

We understand that the term "main outcome" was not adequately explained in the Methods section. We have revised the text to provide a clear definition of the main outcome measure under study.

“Furthermore, studies lacking clear data on meningioma tumors were excluded to ensure the inclusion of high-quality and relevant studies. “ was substituted to “ Furthermore, to enhance the quality and relevance of the study, we excluded research papers that did not provide clear data on whether they were reporting meningioma tumors or not. This exclusion was implemented to ensure that only high-quality studies were included in the analysis.”

#3 Your exclusion criteria are somewhat vague - what is "clear data on meningioma tumors" (Methods section)?

Response: 

We apologize for the vagueness of our exclusion criteria. To clarify, we excluded studies that did not clearly report data on meningioma tumors, to ensure that only high-quality and relevant studies were included in the analysis.

We switched the version “Exclusion criteria encompassed studies where the main outcome was not reported, as well as case reports, letters, comments, and reviews.” to “Studies were excluded from our analysis if they lacked reported data on seizure events. Additionally, we excluded case reports, letters, comments, and reviews to focus solely on studies that provided substantive and comprehensive data”

#4 Your Introduction and Discussion should account for the fact that one of your main research questions has been investigated before. For example,  Delgado Lopez et al (Neurologia 2023) as well as Englot et al (J Neurosurgery 2016), and Sue et al (Acta Neurochir 2015) all found that AEDs do not appear to influence postoperative seizures in meningioma. How does your study compare to those, and what novel aspects does yours add?

Response:

We have added in the 1st paragraph of the discussion by switching the following part “Upon conducting a thorough examination of the data, the study's findings indicate that administering anticonvulsant prophylaxis to meningioma patients without seizures is not universally justified” to “Upon conducting a thorough examination of the data, the study's findings indicate that administering anticonvulsant prophylaxis to meningioma patients without seizures is not universally justified, as previously reported by Delgado Lopez et al. [27] and Englot et al. [6]”

We appreciate your suggestion to discuss how our study compares to previous research on this topic. While studies like Delgado Lopez et al (Neurologia 2023), Englot et al (J Neurosurgery 2016), and Sue et al (Acta Neurochir 2015) have investigated the influence of AEDs on postoperative seizures in meningioma patients, our study builds upon these findings in several key ways:

  1. Our analysis includes seven new studies that were not part of previous investigations of Sue et al and Englot et al..
  2. We performed a pooled analysis of data to provide a comprehensive overview of the incidence of seizure events in the total patient population.

iii. We distinguished between early (within 7 days) and late (>7 days) time periods to gain insight into the timing of seizure occurrences. This distinction is particularly relevant, as Wang et al. (2018) suggested that AEDs might be more effective in preventing early seizures but may have limited efficacy in preventing late events in surgically resected atypical and malignant meningiomas.

#5 Language issues:

We also fixed the previous languages issues reported and revised it again along with the manuscript

By incorporating these revisions, we believe the manuscript will be clearer, more comprehensive, and better positioned to address the reviewers' feedback and contribute to the existing literature on this topic.

Round 2

Reviewer 1 Report

The authors addressed most points adequately.

Author Response

There are no comments to respond to. Please verify!